# Estimating the Impact of Consecutive Blood Meals on Vector Competence of *Aedes albopictus* for Chikungunya Virus

**DOI:** 10.3390/pathogens12060849

**Published:** 2023-06-20

**Authors:** Eva Veronesi, Anca Paslaru, Julia Ettlin, Damiana Ravasi, Eleonora Flacio, Matteo Tanadini, Valeria Guidi

**Affiliations:** 1Institute of Microbiology, Department for Environment Constructions and Design, University of Applied Sciences and Arts of Southern Switzerland (SUPSI), 6850 Mendrisio, Switzerland; damiana.ravasi@supsi.ch (D.R.); eleonora.flacio@supsi.ch (E.F.); valeria.guidi@supsi.ch (V.G.); 2National Centre for Vector Entomology, Institute of Parasitology, Vetsuisse Faculty, University of Zürich (UZH), 5404 Zürich, Switzerland; anca.paslaru@axonlab.com (A.P.);; 3Zurich Data Scientists GmbH, Sihlquai 131, 8005 Zurich, Switzerland; matteo.tanadini@zurich-data-scientists.ch

**Keywords:** fluctuating temperature, infection rate, dissemination rate, transmission rate, transmission efficiency, arbovirus

## Abstract

The continuous expansion of *Aedes albopictus* in Europe and the increases in autochthonous arboviruses transmissions in the region urge a better understanding of the virus transmission dynamic. Recent work described enhanced chikungunya virus (CHIKV) dissemination in *Aedes aegypti* mosquitoes exposed to a virus-free blood meal three days after their infection with CHIKV. Our study investigated the impact of a second blood meal on the vector competence of *Ae. albopictus* from southern Switzerland infected with CHIKV. Seven-day-old *Ae. albopictus* females were exposed to CHIKV-spiked blood and incubated at constant (27 °C) and fluctuating (14–28 °C) temperatures. Four days post-infection (dpi), some of these females were re-fed with a non-infectious blood meal. Virus infectivity, dissemination, transmission rate, and efficiency were investigated at seven and ten dpi. No enhanced dissemination rate was observed among females fed a second time; however, re-fed females have shown higher transmission efficiency than those fed only once after seven days post-infection and incubated under a fluctuating temperature regime. Vector competence for CHIKV was confirmed in *Ae. albopictus* from southern Switzerland. We did not observe an increase in dissemination rates among mosquitoes fed a second time (second blood meal), regardless of the temperature regime.

## 1. Introduction

Arboviruses have become increasingly worrisome in the last decade for several European countries. Chikungunya virus (CHIKV) is responsible for several imported human cases that originated sporadic autochthonous infections and two significant outbreaks in Italy in 2007 and 2017 [1,2]. Moreover, autochthonous human infections with dengue virus (DENV, mainly serotypes 1 and 2) have occurred every year since 2010, with an unprecedented number of cases, including DENV serotype 3, reported in France in 2022 [3]. After the outbreaks of Zika virus (ZIKV) in Central America between 2013 and 2016, the first non-imported transmission in Europe was registered in southern France in 2019 [4]. The autochthonous transmissions of these exotic viruses in continental Europe have been mainly associated with *Aedes albopictus* mosquitoes. Despite the consolidated presence of the vector *Ae*. *albopictus* in southern Switzerland, no local transmissions of mosquito-borne diseases have been here reported, especially in the densely populated urban areas of Canton Ticino [5,6]. However, the number of travellers returning with arboviral infections reported before the COVID-19 pandemic (Federal Office of Public Health, https://www.bag.admin.ch) has risen (accessed on the 15 November 2022), which increases the risk for local transmissions. In 2018, our study on the potential risk for autochthonous transmissions and outbreaks of arboviral diseases in five municipalities of Canton Ticino (Switzerland) revealed that an epidemic could have occurred in the case of the introduction of CHIKV, DENV, or ZIKV during mid-end August [7]. The risk evaluations were determined using data on the vector competence of *Ae. albopictus* available from peer-reviewed papers. However, the vector competence for a virus may vary between different mosquito populations [8,9], and the choice of this value for specific arboviruses can influence the likelihood of an arboviral outbreak. 

For this reason, a better understanding of the capability for arbovirus transmission of *Ae. albopictus* present in Canton Ticino would improve the assessment of threshold values over which an arbovirus transmission may occur, helping in decision-making for timely intervention measures in the occurrence of imported cases. The likelihood of an arboviral outbreak is influenced by several intrinsic and extrinsic mosquito factors, such as feeding behavior, biting rate, population density, survival rate, vector competence of the local mosquito population, and environmental conditions, among others [10]. Environmental conditions, such as seasonal temperature, are known to affect the phenotypic traits of mosquitoes, including vector competence for arboviruses by influencing infection and transmission rates, the gonotrophic mosquito cycle and survival rate, as well as the length of time required for the virus to be transmitted by a vector (extrinsic incubation period, EIP) [11]. Since mosquitoes are exposed to temperature variations in their habitats, vector competence studies should ideally be performed under fluctuating temperature regimes resembling the temperature conditions in the field.

Currently, there is a lack of experimentally proven gold standard methods for vector competence studies with arboviruses [9]. Nevertheless, most experiments are conducted at constant optimal rearing temperatures, which may bias the evaluation of the transmission risks. Since mosquitoes in nature take a blood meal in between each gonotrophic cycle, a single infectious blood meal is commonly provided to mosquitoes when assessing their vector competence under laboratory conditions. However, some species of mosquitoes, such as *Aedes aegypti* [12], usually take frequent blood meals during a reproductive cycle. This natural behavior has been considered in the study of Armstrong and colleagues [13], which interestingly showed how a second non-infectious blood meal decreases the EIP and enhances vector competence for Zika, dengue, and chikungunya viruses in *Ae. aegypti*.

In the present study, we investigated the vector competence of *Ae. albopictus* mosquitoes from the municipality of Muzzano (Canton Ticino, Switzerland) for CHIKV (mutated strain 06.21_E1-A226V, [14]), under a more realistic fluctuating temperature regime. Furthermore, we tested the effect of a consecutive non-infectious blood meal on the infection, dissemination, and transmission for CHIKV 06.21_E1-A226V on the same *Ae. albopictus* population.

## 2. Material and Methods

### 2.1. Mosquitoes Collection

Adult females of *Ae. albopictus* were generated in the laboratory from eggs collected in the field using oviposition traps (ovitraps). A total of 35 ovitraps were placed from May to October 2020 in the municipality of Muzzano, Canton Ticino (Switzerland) [5] and the collected *Ae. albopictus* eggs were sent to the National Center for Vector Entomology (NZVE) at the University of Zürich UZH (Switzerland), where larval stages were reared according to experiment 1 described in [15]. Briefly, two to three days post-egg immersion, hatched larvae were counted and split in trays at a density of 2 larvae per mL (2000 larvae/1000 mL deionized water). During the first four days of the experiment, gradual volumes of 8.3, 16.6, 25, and 33.3 mL of liquid food diet [15] per day were administered to the larvae equal to 0.2 mg/larva/day. All the larvae were incubated at 27 °C with 80% relative humidity (RH). The emerged adults were transferred in polyester cubic netted cages (32.5 × 32.5 × 32.5 cm) (Bugdorm 43030F, MegaViewScienceCo., Ltd., Taichung, Taiwan) and incubated in a climatic chamber with temperature and humidity conditions set as long daylight conditions (16L:8D), including one hour each of dusk and dawn. The adult food source consisted of a 10% sucrose solution replaced every second day. 

### 2.2. Virus

The CHIKV (06.21_E1-A226V isolate, La Reunion Island) [14] strain was used in this study. The isolate was supplied by Anna-Bella Failloux and amplified on an *Ae. albopictus* cell line (C6/36). Briefly, C6/36 cells were grown in sterile cell culture flasks (T25, 50 capacity) and maintained in a Leibovitz‘s medium (Gibco, Thermo Fisher Scientific, Reinach, Switzerland) supplemented with 1% antibiotics-antimycotics (penicillin, streptomycin, and amphotericin B) and 4% fetal bovine serum (FBS). Confluent layers of C6/36 cells were inoculated with 100 µL of the original isolate and incubated at 28 °C w/o CO_2_ for three days. After the incubation, the supernatant from the infected flask was propagated another time in a new flask layered with a C6/36 cell line generating a P2 passage. 

### 2.3. Mosquito Infection

Seven-day-old F1 and F2 mosquito females were starved for 24 h before feeding on heparinized bovine blood (obtained from the cattle clinic of the Veterinary Faculty, Zürich) spiked with CHIKV, as earlier described [16], with the exception that we used an artificial membrane (Parafilm M, Sigma-Aldrich, Buchs, Switzerland) for the Hemotek feeders (Hemotek Ltd., Lancashire, UK) instead of a pork intestine membrane. To increase the mosquitoes’ attraction to the feeder and augment feeding rates, women’s ankle nylon socks (Coop, Zürich) worn overnight were added on top of the Hemotek feeders. In addition, to improve the feeding rate, a box containing dry ice simulating the exhalation of CO_2_ was placed close to the feeders. Blood-fed mosquitoes were sorted as described [17]. Immediately after each feeding, one engorged female/feeding was collected as a day 0 sample to serve as baseline data to confirm infection and stored at −80 °C until further examinations. 

All the remaining engorged females were transferred into netted cardboard pots (maximum 80 females/pot) and split into (i) a control group (receiving one infectious blood meal only) and (ii) a second blood meal group (receiving an additional non-infectious blood meal four days after the first one). The control group females were incubated for seven and ten days with 10% sucrose as a food source, which was replaced daily. To improve the feeding rate, mosquitoes from the second blood meal group were induced to lay eggs using a modified method described by [18]. The bottom of the cardboard box cylinder [16] was modified in three layers from top to bottom: a fine layer of black women’s ankle nylon socks, egg-laying paper, and a fine net. On day four post-infection, a further (non-infectious) blood meal was offered to the second blood meal group. Engorged females were selected and incubated in a new cardboard box for an additional three and six days with the 10% sucrose solution as described above. Two different climatic conditions were applied for the incubation of both groups: i) constant temperature (CT) (27 °C and relative humidity 85%), and ii) daily fluctuating temperature (FT) (14–28 °C with a mean of 23 °C and 60–85% relative humidity) (Figure 1). 

At each specific time point, 20–30 live females from each group were collected and prepared for virus detection and quantification as described below [16]. 

The exposure of mosquitoes to the infectious blood, the selection of engorged females, and further incubation were carried out in a biosafety containment level 3 (BSL3) laboratory of the Laboratory Animal Service Centre (LASC) at the UZH.

### 2.4. Mosquito Dissection and Saliva Collection

After each incubation point, all the surviving mosquitoes were immobilized on a chilled table, had their legs and wings removed, body parts and saliva collected, and were finally homogenized, as described in [17]. 

Saliva collection and virus quantification were completed as in [16]. Briefly, each proboscis was inserted into a 20 µL pipette tip filled with 5 µL of FBS. After 30 min of “blind” salivation, the 5 µL of FBS was flushed into a 1.5 mL Eppendorf tube containing 45 µL of Dulbecco’s Modified Eagle Medium (DMEM) (Gibco, Thermo Fisher Scientific, Reinach, Switzerland) supplemented with 1% antibiotics and fungizone (1000 IU/mL penicillin/streptomycin; 4 µg/mL amphotericin) (Gibco, Thermo Fisher Scientific, Reinach, Switzerland) (DMEM complete), giving a final volume of 50 µL. All the saliva samples were stored at −80 °C until further analyses. The 6-well plates layered with 75–80% confluent Vero cells (800,000 cells/2 mL/well) were incubated for 24 h with DMEM complete supplemented with 10% FBS. For the inoculation, the medium in each well was removed and replaced with 295 µL of DMEM supplemented with 2% FBS, and finally, 5 µL of the saliva sample was inoculated. After 1 h incubation at 37 °C, 4 mL of a 2% agarose solution (UltraPure™ Agarose, Invitrogen Life Technologies, Renfrew, UK) in DMEM complete was added to each well without removing the saliva inoculum, and all plates were incubated at 37 °C with 5% CO_2_. After three days of incubation, the agarose gel was removed, and the cells were stained with 1–2 mL of crystal violet solution per well. After 30 min incubation at room temperature, the cells were rinsed with water, and each well was checked for the presence of plaques.

### 2.5. Quantification of Virus Infection, Dissemination, and Transmission

Infection, dissemination, and transmission were investigated by the quantification of viral RNA from the body, legs and wings, and saliva, respectively. Briefly, viral RNA was extracted from tissues’ homogenate using the RNA mini kit (Qiagen, Hilden, Germany) following the manufacturer’s instructions (elution volume 60 μL) and quantified by a reverse transcription-quantitative polymerase chain reaction (RT-qPCR), using the Power SYBR™ Green RNA-to-CT™ 1-Step Kit (Applied Biosystems™, Thermo Fischer, Zug, Switzerland) and the following primers: 10 µM Chik/E2/9018/+ (CACCGCCGCAACTACCG) and anti-sense10 µM Chik/E2/9235/- (GATTGGTGACCGCGGCA) [14]. The amplification was carried out using 2 µL of RNA and performed in a QuantStudio™ 7 Flex (Applied Biosystems, Thermo Fischer, Zug, Switzerland). Each reaction was run in duplicate following the program provided by Dr. Anna-Bella Failloux and Marie Vazeille (Institute Pasteur, Paris, France): reverse transcription step at 48 °C for 30 min, inactivation step of RT/RNAse enzyme at 95 °C for 10 min followed by 40 cycles of 95 °C for 15 s, 60 °C for 1 min, 72 °C for 30 s, and a final step at 95 °C for 20 s. The specificity of the PCR products was secured with a melting curve analysis after amplification. Finally, a standard curve of the virus was generated by using duplicates of the 10-fold serial dilutions of the viral inoculum used to feed the mosquitoes and converting the viral RNA cycle threshold (Cq) values into a plaque-forming unit (PFU) as described [18]. 

### 2.6. Data Analysis

The vector competence was described through five indexes: infection rate (IR, proportion of females with infected abdomen among tested ones), dissemination rate (DR, proportion of females with infected legs and wings among the infected ones), transmission rate (TR, proportion of females with infected saliva among the ones with disseminated infection), dissemination efficiency (DE, proportion of females with infected legs and wings among all tested ones), and transmission efficiency (TE, proportion of females with infectious saliva among all tested ones). 

All rates and efficiencies computed for infection, dissemination, and transmission were analyzed as binomial data. The Wilson method was used to estimate 95% confidence intervals for the binomial proportions.

The plaque-forming unit data were analyzed with a generalized linear model with a family set to quasi-Poisson to account for overdispersion. The starting model contained the three-fold interaction among the days post-infection, temperature regime (constant or fluctuating), and experiment type (control or second blood meal). The significance level was set at 5%. All the statistical analyses were conducted using the statistical software R (version 4.2.0) [19].

For further details related to the statistical analyses refer to the Appendix.

## 3. Results

### 3.1. Virus 

The CHIKV isolate (06.21_E1-A226V) was propagated twice on C6/36 cells (P2) and reached a final titer of 7.75 log_10_TCID_50_/mL. Based on the standard curve of P2 obtained, a cut-off of Cq ≤ 33.7 was applied to the RT-qPCR results. 

### 3.2. Mosquito Infection

A total of 1336 *Ae. albopictus* mosquitoes were exposed to washed heparinized blood mixed (1:10) with the CHIKV P2 passage, yielding a final titer of 6.75 log_10_TCID_50_/mL. 

Overall, 843 (63%) were fully engorged and were split as follows: 7 mosquitoes were kept as day 0 specimens, 205 for the control group (83 at CT and 122 at FT), and 631 (326 at CT and 305 at FT) for the second blood meal group which was exposed to a non-infectious blood meal 4 days post-infection (dpi). From the 326 incubated at CT, 300 survived (92%), and 98 (33%) were fully engorged when exposed to the second blood meal. In contrast, from the 305 mosquitoes incubated at FT, 268 (88%) survived and were exposed to a second blood meal, of which 94 (35%) were fully engorged (Appendix A).

### 3.3. Virus Amplification, Group Treatment, and Temperature Conditions

Overall, mosquitoes fed a second time (“second blood meal group”) and incubated under a CT regime, consistently displayed lower infection, dissemination, and transmission rates and efficiencies compared to mosquitoes fed only once (control group) (Table 1, Figure 2). 

This pattern was consistent across 11 out of 12 tests (p-value 0.63%, binomial test) when incubated at CT, except for the second blood meal group reaching a higher dissemination rate than the control group at seven days post-infection (Figure 2).

A significant difference in infection rate was recorded in the control group, with values ranging between 74 and 96% and 65 and 94% on day 7 and 10 post-infection, respectively, whereas the second blood meal group had a lower range of 15–52% and 18–51% at 7 and 10 dpi, respectively (Table 1, Figure 2). 

We did not observe a clearly distinguished pattern between the two group treatments when the incubation was performed under a fluctuating temperature regime (Figure 3). Indeed, half of the samples from the second blood meal group (regardless of the type of infection and time post-oral infection) had higher efficiency rates than the control group. Specifically, on day seven, dissemination and transmission rates and efficiencies from the second blood meal group were slightly higher than the control one, whereas the infection rate was higher for the control group on both days post-infection.

### 3.4. Infectious Virus Particles

When incubated under the CT regime, infectious virus particles detected in the saliva (by the means of the plaque-forming unit—PFU) of positive females were higher in the control group than in females from the second blood meal group (Figure 4), either 7 dpi or 10 dpi. A PFU decrease was observed in both groups with time.

Under the FT regime, the number of infectious virus particles among the second blood meal group incubated for seven days was higher than in the control females. This discrepancy was reduced with time as the PFU values of the former group decreased after 10 days to have similar values as the control females. 

## 4. Discussion

With this study, we demonstrated the potential role of *Ae. albopictus* from Muzzano (Canton Ticino, Switzerland) in the transmission of CHIKV (strain 06.21, E1-A226V). A consecutive blood meal did not enhance the virus infection, dissemination, and transmission of CHIKV for *Ae. albopictus*. 

Our data on the transmission efficiencies (TE) of mosquitoes that did not feed a second time (control) at a constant temperature (CT, 28–38%) do not differ much from the work of Fortuna and colleagues [20] with a TE of 41% for *Ae. albopictus* derived from a long-established colony in Italy (Scalea, Calabria). Previous studies investigating vector competence for strain 06.21, E1-A226V among two populations of *Ae. albopictus* from Canton Ticino, reported a TE of 8% (Arogno population) and 54% (Tenero population) [8] after seven days of incubation at a CT of 28 °C. This highlights how distinct populations of the same mosquito species may differ in their ability to transmit the same virus strain. This is mainly due to the different effectiveness of the mosquito midgut and salivary gland barriers in preventing the escape or entry of the infectious virus particles in the haemocoel and salivary glands, respectively, and how the virus has to overcome different antiviral immune responses that can limit virus infection, resulting in variations in vector competence between distinct populations [10]. 

Our results are in contrast with the conclusions of Armstrong et al. [13] about enhanced virus dissemination from the midgut to the haemocoel of *Ae. albopictus* infected with Zika virus when further exposed to a second non-infectious blood meal. The administration of a second non-infectious blood meal did not increase the vector competence of *Ae. albopictus* mosquitoes previously infected with CHIKV. Overall, the infection, dissemination, and transmission rates of mosquitoes exposed to a second non-infected blood meal and incubated at CT were consistently lower than those recorded among mosquitoes fed only once as evinced in 11 trials out of 12 (*p*-value 0.63%). No statistically significant differences were observed within the two groups, except for the infection rate for females fed only once and incubated at CT, showing values significantly higher than those fed a second time. This trend was not so clear at FT, where we did not see a significant increase for one of the two groups. 

However, a peculiar trend was observed for the virus dissemination within the vector’s body concerning the incubation temperature regime. Seven days post-infection, females from the second blood meal group display a large discrepancy between IR and DR percentages when incubated at CT. Indeed, an IR of 15–52% and a DR of 61–100% suggest that most of the infectious virus particles are no longer present in the midgut but have disseminated into the haemocoel, bypassing the escape midgut barrier. The propagation of infectious virus particles from the first site of infection (midgut) to the salivary glands is controlled by a series of barriers that prevent the virus from entering the midgut cells (infection barriers), escaping from them after its amplification (escape barriers) to disseminate in the whole haemocoel (dissemination barriers), and finally reach the salivary glands (salivary gland barriers). Indeed, the integrity of the basal lamina separating the midgut cells from the haemocoel is crucial for preventing the escape of the virus particles from the gut lumen to the haemocoel. The work of Armstrong et al. has shown clear damages in the basal lamina of mosquitoes fed a second time post-virus infection [13], with microperforations related to enzymatic degradation [21], or most likely to a mechanical distention of the lamina as damages have been observed immediately after blood engorgement [13]. While our results do not seem to follow Armstrong’s conclusion, we could speculate that, however, the stress to which the basal lamina is exposed during a second blood meal could be related to the amount of blood that is ingested. Even though we only collected fully engorged females in both the first and second blood meals, it is difficult to perceive a difference in the volume of ingested blood if it is in the order of microliters unless it is measured. Considering that a female’s intake is approximately 2–4 µL of blood [22,23], an imperceptible variation of ingested blood can actually have a higher impact on mosquitoes. This could perhaps explain why at day 7 post-infection, the percentage of females incubated at CT (27 °C) have a lower infection rate (15–52%) in the second blood meal group compared to the control group (74–96%), whereas, at FT, the percentages of the two groups are similar (44–77% and 52–85% for the second blood meal and control group, respectively). Indeed, high temperatures could imply faster digestion [24,25]; therefore, females become more prone to feed a second time if exposed to another blood meal, as they cleared the previous blood from the midgut, leading to damage of the basal lamina due to the larger volume of blood ingested. On the contrary, at lower mean temperatures (as in the fluctuating temperature regime), they are less prone to feed a second time, or they ingest a smaller volume of blood compared to the ones kept at higher temperatures; therefore, there should not be microperforations, and virus particle dissemination into the haemocoel is less due to the midgut barriers. This may also explain why at a higher temperature, the discrepancy between the percentage of females with an infection is larger than those with dissemination after seven days post-infection, since the virus particles are already escaped from the midgut and are now in the haemocoel. These are hypotheses that should be further investigated.

Arthropods do not thermoregulate; hence, the atmospheric temperature substantially impacts the virus’s amplification and dissemination within their body, affecting the length of time required by the pathogens to infect, disseminate, and be transmitted (extrinsic incubation period—EIP). 

A similar study by Stephenson et al. [9] on the potential transmission of DENV-4 for *Ae. aegypti* mosquitoes showed that the infection rates of midgut and saliva samples of mosquitoes receiving one blood meal did not differ from those of mosquitoes receiving a second non-infectious meal. This indicates that additional blood meals do not impact the DENV-4 transmission potential or transovarial transmission. Although we initially designed our experiment to be similar to the one from Armstrong et al. with the second exposure at three days post-infection, the feeding rate of *Ae. albopictus* to the non-infectious blood was very low or zero, probably due to the impact of the temperature on the gonotrophic cycle [26]. Moreover, the strain of *Ae. albopictus* they used was from laboratory-reared species and already adapted to feed multiple times in between two gonotrophic cycles. The choice of using field-derived mosquitoes or mosquitoes that were reared for many generations and thus possibly adapted to laboratory conditions may influence the outcome of vector competence studies [10]. 

The implication of arbovirus transmission due to possible damage of the midgut basal lamina has also been discussed by Balestrino et al. [18]. The authors have measured the transmission rate and viral load among pools of irradiated *Ae. albopictus* and *Ae. aegypti* species after infection with the CHIKV mutant strain (06.21 E1-A226V) and DENV-2. An increase in the virus in the body of mosquitoes infected with CHIKV was observed for both species. Still, they did not find an increase in the transmission rate for both viruses in the two species, except a slight marginal increase at the individual level for DENV-2-infected *Ae. albopictus*. The incubation temperature regime used in this work was 27 °C constant. This suggests that salivary gland barriers worked more efficiently than midgut barriers, inferring damage to the basal lamina.

More studies are needed to investigate the dynamics of virus dissemination concerning damages in the basal lamina and how temperature can impact on the degeneration and regeneration process. Very little is still known about the temperature conditions for mosquitoes during their resting time [27]. Females only fly when searching for sugar or blood, so they spend most of their time in the shade. Currently, there is a lack of experimentally proven gold standard methods for vector competence studies for arboviruses, making comparisons difficult [9]. As already highlighted, several intrinsic and extrinsic factors may affect the biology of mosquitoes, and, consequently, their ability to transmit a virus. In laboratory studies, the choice of the viral strain, the method of virus stock production (such as the passage number, cell type, and multiplicity of infection), the viral infectious dose, the blood feeding method, the blood source, temperature, and humidity conditions may impact the virus dynamic and affect the vector competence of the mosquitoes. 

## 5. Conclusions

This study provides evidence of the potential role of *Ae. albopictus* from southern Switzerland (Muzzano, Canton Ticino) in the transmission of chikungunya virus. A second feeding with non-infected blood did not enhance CHIKV dissemination in females of this population. We here emphasize the importance of temperature conditions during the incubation of infected mosquitoes when designing vector competence studies, as this can significantly impact the outcome of the results. The lack of data on the ecology of mosquitoes in relation to the temperatures at their resting sites needs to be taken into consideration, and more studies on this subject are needed.

## Figures and Tables

**Figure 1 pathogens-12-00849-f001:**
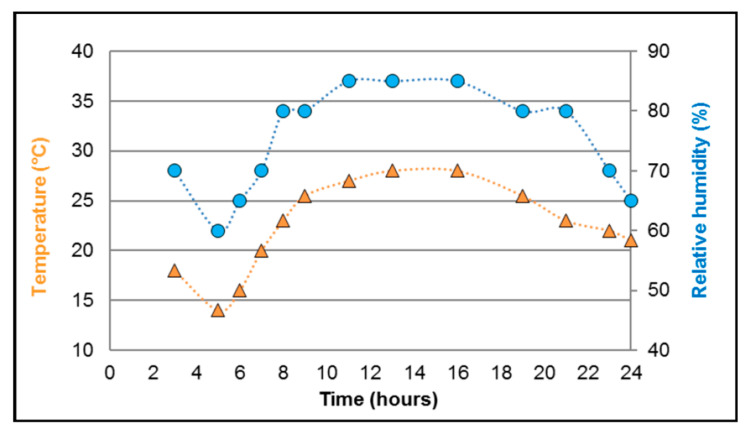
Daily fluctuating temperature and relative humidity regimes used in this study.

**Figure 2 pathogens-12-00849-f002:**
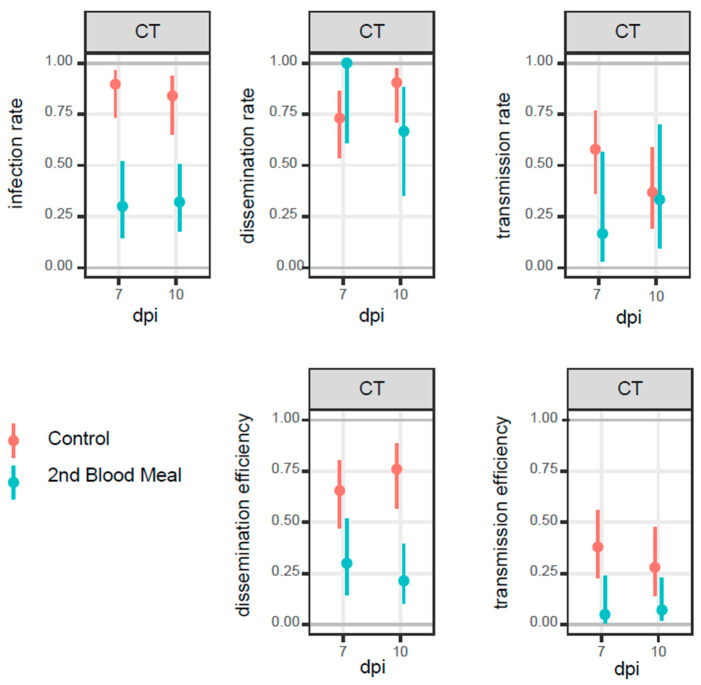
Infection rates (IR), dissemination rates (DR), transmission rates (TR), dissemination efficiencies (DE), and transmission efficiencies (TE) from *Ae. albopictus* infected with CHIKV 06.21_E1-A226V for the control group (orange bars) and the second blood meal group (light blue bars) at days 7 and 10 post-infection (dpi) under constant (CT) regime. Error bars: 95% confidence intervals.

**Figure 3 pathogens-12-00849-f003:**
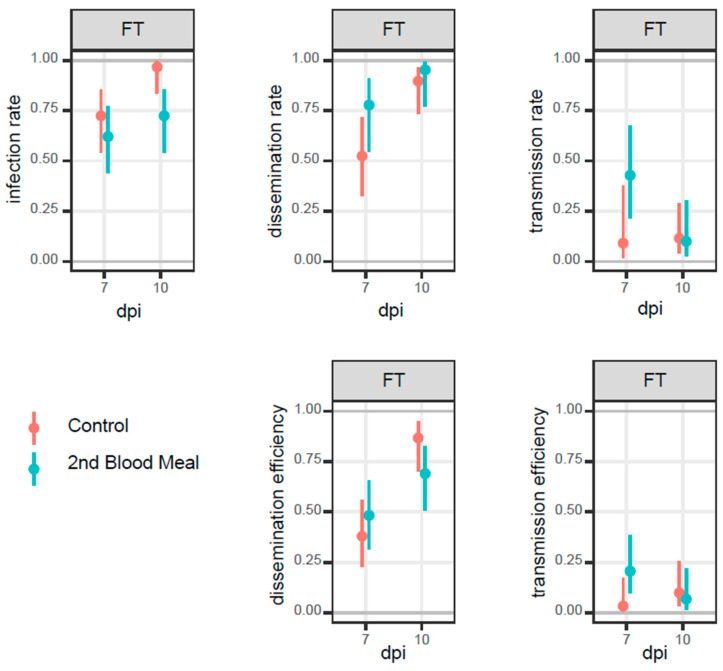
Infection rates (IR), dissemination rates (DR), transmission rates (TR), dissemination efficiency (DE), and transmission efficiencies (TE) from *Ae. albopictus* infected with CHIKV 06.21_E1-A226V for the control group (orange bars) and second blood meal group (light blue bars) at days 7 and 10 post-infection (dpi) under fluctuating (FT) regime. Error bars: 95% confidence intervals.

**Figure 4 pathogens-12-00849-f004:**
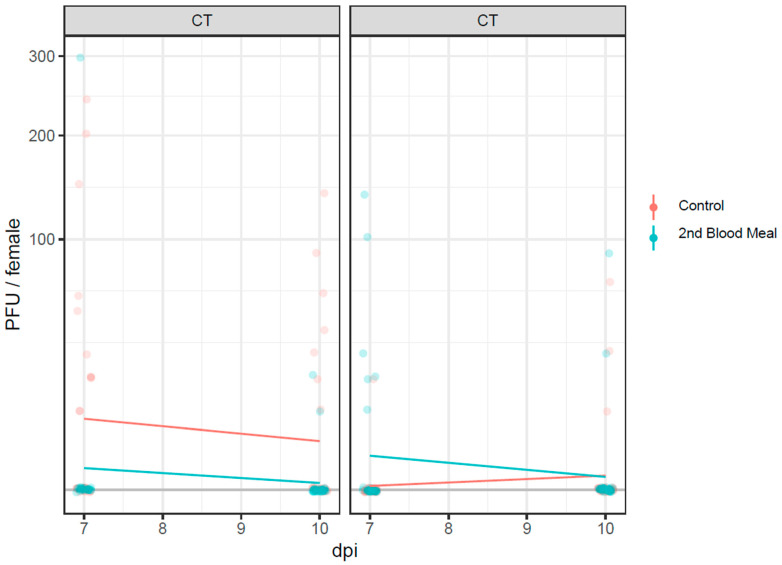
Infectious virus particles (PFU) measured in the saliva of females from the control and second blood meal group after incubation at constant (CT) and fluctuating (FT) temperature conditions. Orange dots and bars: control group; light blue dots and bars: second blood meal group.

**Table 1 pathogens-12-00849-t001:** Rates of infection (IR), dissemination (DR), transmission (TR), and efficiency of dissemination (DE) and transmission (TE) from *Ae. albopictus* infected with CHIKV 06.21_E1-A226V for the two groups (control group and second blood meal group) at both constant (CT) and fluctuating (FT) temperature regimes at 7 and 10 days post-infection (dpi). The ranges of PFU measured (min-max) for saliva in the two groups and two temperature sets are also reported.

Group	dpi	IR (%)				DR (%)						TR (%)		DE (%)				TE (%)					
		CT (%)	CI95 (%)	FT (%)	CI95 (%)	CT (%)	CI95 (%)	FT (%)	CI95 (%)	CT (%)	CI95 (%)	FT (%)	CI95 (%)	CT (%)	CI95 %	FT (%)	CI95 (%)	CT (%)	CI95 (%)	PFU (min-max)	FT (%)	CI95 (%)	PFU (min-max)
**Control**	7	26/29 (90)	74–96	21/29 (72)	52–85	19/29 (66)	47–80	11/29 (38)	23–56	11/29 (38)	23–56	1/29 (3)	1–17	19/26 (73)	54–86	11/21 (52)	32–72	11/19 (58)	36–77	10–240	1/11 (9)	2–38	0–20
	10	21/25 (84)	65–94	29/30 (97)	83–99	19/25 (76)	57–89	26/30 (87)	70–95	7/25 (28)	14–48	3/30 (10)	3–26	19/21 (90)	71–97	26/29 (90)	75–96	7/19 (37)	19–56	10–140	3/26 (12)	4–29	10–70
**Second blood meal**	7	6/20 (30)	15–52	18/29 (62)	44–77	6/20 (30)	15–52	14/29 (48)	31–66	1/20 (5)	1–24	6/29 (21)	10–38	6/6 (100)	61–100	14/18 (78)	55–91	1/6 (17)	3–56	0–300	6/14 (43)	21–67	10–140
	10	9/28 (32)	18–51	21/29 (72)	54–85	6/28 (21)	10–40	20/29 (69)	51–83	2/28 (7)	2–23	2/29 (7)	2–22	6/9 (67)	35–88	20/21 (100)	77–99	2/6 (33)	10–70	10–20	2/20 (10)	3–30	30–90

## Data Availability

All data generated or analyzed during this study are included in this published article and its Appendix A.

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
