# Peer review of "Estimating the Impact of Consecutive Blood Meals on Vector Competence of Aedes albopictus for Chikungunya Virus"

_pathogens, 2023, doi:10.3390/pathogens12060849_

Round 1

Reviewer 1 Report

The manuscript is interesting and presents util data. Nevertheless, I have some concerns about the presentation and analysis of the results. Authors talk about "consistent" differences between control and 2 time fed females. But, for me, the differences are ,ore inconsistent than consistent, because of the lack of statistically significancy and sustained tendency of the results. The analysis is not always clear and, also, it can be repetitive from one section to another. The lack of statisical tests representation in the figures and text is a problem. I have some punctual recommendations made in the manuscript.

Author Response

Dear reviewer

Thank you for your suggestions and comments. We have now amended the manuscript. Please find here the replies to your comments.

We have now also submitted an Appendix to our manuscript with all the statistical analyses. It was meant to be sent with the first submission, but for some reason, it was not. Apologises. 

Thank you for your time.

Best regards

Eva

Reviewer 2 Report

The authors of this paper conducted a study on the impact of a second blood meal on the vector competence of Ae. albopictus infected with CHIKV in southern Switzerland. The manuscript is well-organized and well-written, following scientific logic.

 Major comments: It is interesting to note that there are lower infection and dissemination rates at a constant temperature with a second blood meal. However, there is no difference in fluctuating temperatures, as shown in Figure 2. However, in Figure 4, the infection increases when the authors evaluate the PFU/female after a second meal at fluctuating temperatures. I wonder if the authors used all the infected mosquitoes (infected in the midgut or wings and without dissemination in the salivary gland) or only those with a positive infection in the salivary gland in Figure 4.

The low infection levels in mosquitoes may not be significant. Is it common to find a low PFU in the salivary glands of mosquitoes infected with chikungunya?

In terms of minor comments, I would suggest standardizing the color of the latter in Lines 153-155 for consistency purposes.

Author Response

Dear reviewer

Thank you for your suggestions and comments. We have now amended the manuscript. Please find here the replies to your comments.

Thank you for your time.

Best regards

Eva

Reviewer 3 Report

This paper shows two interesting pieces of data: the role of temperature in CHIKV dissemination in a population of Ae. albopictus from Europe (where CHIKV keeps expanding) and the role of a second bloodmeal in the success of the virus within the mosquito vector. This is an important point, as studies looking to replicate this second bloodmeal phenomenon have not been as numerous they should. The paper has important data and will be impactful in the field, but there are some data display issues that should be addressed prior to publication.

Overall:

Instead of talking about “temperature” it would be more appropriate to talk about something more like “temperature profile type” because you did not compare different temperatures per se, just two types of profiles (constant vs. fluctuating).

Methods: 

2.2: Why was the virus P2 from C6/36 used rather than passaging in mammalian cell line?

2.3: The women’s nylon trick wasn’t immediately apparent in the reference [17] – is there data to support this works? If so, would be worth a supplemental figure for future citations.

Was time put into the model as a continuous variable? If so, this is likely inappropriate as only 3 times are included and they’re relatively arbitrarily chosen days and we do not know if only 3 days is significant enough time to constitute a significant timestep to capture this phenomenon (if it exists). It would be more appropriate to think of this as a categorical variable of dpi.

Results:

Is “second group” and “second bloodmeal” the same thing?

What is the difference between Table 1 and Table 2?

Figures 2 & 3: I’m confused as to what each panel represents… 

3.4: saying there is no significant “time” effect is a bit overstating because only 2 arbitrary days were tested. It would be more appropriate to say that no difference exists between 7 and 10 dpi (see comment above).

For another example of how a second bloodmeal does not affect vector competence, see Mayton et al. 2020 Parasites & Vectors.

Author Response

Dear Reviewer

Thank you for your feedback and useful comments and suggestions.

We have now amended the manuscript, please find here the replies to your questions.

Thank you for your time.

Best regards

Eva Veronesi
